# Tai Chi Exercise for Mental and Physical Well-Being in Patients with Depressive Symptoms: A Systematic Review and Meta-Analysis

**DOI:** 10.3390/ijerph20042828

**Published:** 2023-02-05

**Authors:** Norliyana Abdullah Sani, Siti Suhaila Mohd Yusoff, Mohd Noor Norhayati, Aida Maziha Zainudin

**Affiliations:** 1Department of Family Medicine, School of Medical Sciences, Universiti Sains Malaysia, Kubang Kerian 16150, Kelantan, Malaysia; 2Department of Pharmacology, School of Medical Sciences, Universiti Sains Malaysia, Kubang Kerian 16150, Kelantan, Malaysia

**Keywords:** Tai chi, mental well-being, physical well-being, depressive, randomized control trial

## Abstract

Tai Chi is a mindfulness–body practice that has physiological and psychosocial benefits and can be integrated into the prevention and rehabilitation of various medical conditions; however, the effectiveness of Tai Chi in the treatment of depression remains unclear. This review aimed to determine the effects of Tai Chi exercise on mental and physical well-being in patients with depressive symptoms. We searched databases for English language publications that appeared during January 2000–2022. The included trials were RCTs that involved people with depression with no other medical conditions, and included both adolescent and adult samples. A meta-analysis was performed using a random effects model and the heterogeneity was estimated using I^2^ statistics. The quality of each trial was assessed according to the Grades of Recommendation, Assessment, Development, and Evaluation (GRADE) methodology. The eight trials were divided into two comparisons: (1) a combination of Tai Chi and antidepressants versus standard antidepressants; (2) Tai Chi versus no intervention. The Tai Chi intervention showed improvements in mental and physical well-being as evidenced by the reductions in depression and anxiety and improved quality of life (QOL) of the patients with depressive symptoms. Further well-controlled RCTs are recommended with a precision trial design and larger sample sizes.

## 1. Introduction

Depression is a common mental illness, with an estimated 3.8% of the worldwide population being affected [1]. It is the second leading cause of years lived with disability globally, at up to 10.8% [2]. The emergence of the COVID-19 pandemic in 2020 saw a global increase in psychological problems due to economic and social burdens. A recent systematic review reported that increasing rates of COVID-19 infection, combined with associated factors, had caused an increase in the prevalence of major depressive disorder [3]. There was an increased number of children and adolescents displaying depressive symptoms during the COVID-19 pandemic [4,5]. According to the National Health and Morbidity Survey 2019, an estimated 2.3% of Malaysia’s adult population, or half a million people, suffered from depression [6]. The prevalence rates of depression in Malaysia range from 3.9% to 46% [7]. According to the Diagnostic and Statistical Manual of Mental Disorders, 5th edition (DSM-V), this disorder can present with depressed mood, loss of interest or pleasure, decreased energy, feelings of guilt or low self-worth, disturbed sleep or appetite, and poor concentration. Depression often comes with symptoms of anxiety and mood disturbance. These problems can become chronic or recurrent and can lead to significant impairments in an individual’s quality of life (QOL) and self-care, which can eventually lead to the risk of chronic illnesses such as coronary heart disease or cancer or a risk of death due to suicide [8,9].

Antidepressant therapies, mainly selective serotonin receptor inhibitors, have been used for the treatment of depression. There has been a significant increase in the usage of antidepressants to treat depression, anxiety, and adjustment disorders, whereby the usage percentages of concomitant antipsychotic medication have also increased, with a reduction in percentages for those who underwent psychotherapy, reflecting the greater focus on pharmacological treatments more than the psychological dimension of care [10]. In a single-centered observational study, it was stated that polypharmacy in the concomitant use of two antidepressants was as frequent as monotherapy [11]. Hence, this will expose patients to more adverse effects of antidepressants. Our concern regarding the wide use of antidepressants is because this will cause adverse effects such as an increased risk of suicidal ideation and behavior, especially in youth groups [10,12]. Another study revealed that many antidepressants available on the market are associated with sexual dysfunction and sleep disturbances, which are some of the most common reasons for non-adherence toward antidepressants [13,14]. According to at least one study, patients with depression can experience unwelcome side effects rather than structural barriers or non-adherence behaviors, leading to non-adherence with antidepressants [15]. A review of the non-pharmacological treatment of depression through physical exercise therapies, including Tai Chi, reported lower side effects [16] and lower relapse rates [17], making it a cost-effective [18] approach for the treatment of depression.

A meta-analysis also concluded that antidepressant use is beneficial in severe depression, although the effects may be minimal or non-existent in mild and moderate depression [19]. Hence, non-pharmacological interventions may play a role in those with milder forms of depression. A systematic review of non-pharmacological treatments of depression, including physical activity interventions such as Tai Chi, Qigong, and yoga, reported a reduction in depressive symptoms, meaning they may be used as adjuncts to antidepressant therapy [16].

Tai Chi, also called Taiji or Tai Chi Chuan, is a form of mindfulness exercise that originated and remains widely practiced in China. It combines Chinese martial arts and meditative movements that promote balance and the healing of the mind and body, involving a series of slowly performed dance-like postures that flow into one another [20]. As it combines mental concentration, physical balance, muscle relaxation, and relaxed breathing, Tai Chi shows excellent potential to be integrated into the prevention and rehabilitation of medical and psychological conditions [21,22]. Tai Chi is an exercise that modulates the activity and connectivity of key brain regions involved in depression and mood regulation, reducing neuroinflammatory sensitization and modulating the autonomic nervous system, which improves emotion regulation and reduces stress [23].

Several trials have reported the effectiveness of Tai Chi exercise as part of the non-pharmacological approach to treat patients with depression, and it has been associated with improvements in mental health and QOL among the healthy general population [24,25,26]. Tai Chi has also been shown to have significant impacts on depression, anxiety, and physical well-being among people with a variety of chronic conditions [27,28,29]. A meta-analysis also showed reductions in depression with a Tai Chi intervention in an elderly population [30,31]. A limited number of trials have studied the effects of Tai Chi on depression symptoms among the younger generation, probably due to mindfulness exercise not being attractive to them. Tai Chi exercise also leads to a better QOL by improving the physical functional status [32], insomnia [33], and chronic pain [34]—conditions that are common among people with depression. A few systematic reviews have reported reductions in medical-related symptoms in people with cardiovascular-related illness [31,35,36], fibromyalgia [37], and multiple sclerosis [38]. Those reviews assessed the effects of Tai Chi when aiming to reduce symptoms of medical conditions and depressive symptoms; hence, their results were more complex and unclear. Several studies have assessed depressive symptoms as secondary outcomes; thus, the included participants can be participants without baseline depression. Most trials examine depressive symptoms as secondary outcomes of the research, and the participants involved have underlying medical conditions that can be debilitating physically and can lead to chronic pain, which can influence the severity of the depressive outcome. The physiological and psychosocial effects of Tai Chi are beneficial in the prevention and rehabilitation of various medical conditions; however, the effectiveness of Tai Chi in the treatment of depression among patients with depressive symptoms and with no other medical conditions remains unclear. This present review aimed to overcome some limitations of the existing reviews by analyzing the effects of Tai Chi exercise on mental and physical well-being in patients with depressive symptoms and no other specific medical conditions.

## 2. Materials and Methods

### 2.1. Search Strategy

We conducted this systematic review according to the protocol previously registered under PROSPERO (registration number CRD42022309924); the meta-analysis was conducted according to the PRISMA guidelines. We searched the Cochrane Central Register of Controlled Trials (CENTRAL) and MEDLINE (Pubmed) published from 1 January 2000 to 1 January 2022. The databases were searched based on the search terms ‘‘Tai Chi, ’’ ‘‘Tai Chi chuan,’’ ‘‘Taiji,’’ ‘‘depression,’’ and ‘‘depressive’’, with Boolean operators of AND and OR. The researchers also searched for ongoing trials through the World Health Organization International Clinical Trials Registry Platform (ICTRP) and ClinicalTrials.gov. Only English language publications were considered for this review.

### 2.2. Study Eligibility

The criteria considered to be included in this review were based on the PICOS format, namely the population, intervention, comparisons, outcome, and study design. The included trials were randomized controlled trials (RCTs) that compared any style of Tai Chi exercise to the standard treatment or as an adjunct to the standard treatment, no treatment, or health education. Trials with no explicit study design or method, observational studies, therapy guidelines, and reviews were all excluded.

Participants that reported depressive symptoms based on subjective or objective assessments, regardless of age, gender, and ethnicity, were included. Participants with medical illness were excluded in this review. Trials that included Tai Chi as an intervention or co-intervention to other mindfulness exercises were excluded.

This review included trials that reported our primary outcome with or without secondary outcomes. The primary outcome examined in this review was depressive symptoms. The secondary outcomes were anxiety and QOL. The follow-up period following the intervention was at least eight weeks.

### 2.3. Trial Selection

The review authors (N.A.S., S.S.M.Y.) scanned the titles and abstracts from the searches and obtained the full texts of articles that appeared to meet the eligibility criteria or when there was insufficient information to assess the eligibility. These authors independently assessed the eligibility of the trials and documented the reasons for exclusion. Disagreements between the review authors were resolved by discussion.

### 2.4. Data Extraction and Management

The data of interest were extracted by two authors (N.A.S., S.S.M.Y.) from each selected trial and consisted of the following items: study setting, participant characteristics (age, sex, ethnicity), methodology (number of participants randomized and analyzed, duration of follow-up), therapy duration, method for diagnosing depressive symptoms, and outcome measures (severity of depressive symptoms, anxiety, and QOL). Any disagreements were resolved by discussion.

### 2.5. Assessment of Risk of Bias in Included Trials

We assessed the risk of bias based on random sequence generation, allocation concealment, the blinding of participants and personnel, the blinding of outcome assessors, the completeness of outcome data, the selectivity of outcome reporting, and other biases [39]. Any disagreements were resolved by discussion.

### 2.6. Statistical Analysis

We measured the treatment effect for dichotomous outcomes using risk ratios (RRs) and absolute risk reduction; for continuous outcomes, we used standard mean differences (SMDs); all were assessed with 95% confidence intervals (CIs). Forest plots were drawn for the trials with continuous outcomes using mean differences (MDs) and 95% CIs. We checked the included trials for unit of analysis errors, which can occur when trials randomize participants into intervention and control groups in clusters but analyze the results using the total number of participants. We adjusted the results from trials showing unit of analysis errors based on the mean cluster size and intracluster correlation coefficient [39]. We contacted the original trial authors to request missing or inadequately reported data and carried out analyses on the available data in the event that the missing data were not available.

We assessed the presence of heterogeneity in two steps. First, we assessed obvious heterogeneity at face value by comparing populations, settings, interventions, and outcomes. Second, we assessed statistical heterogeneity by means of the I^2^ statistic [39]. If there were sufficient trials, we used funnel plots to assess the possibility of reporting biases, small study biases, or both. Meta-analyses were performed using Review Manager 5.4 software (RevMan 2020, Grantsville, UT, USA). We used a random effects model to pool the data. Thresholds for interpreting the I^2^ statistic can be misleading, since the importance of inconsistency depends on several factors. We used the guide to interpret heterogeneity as outlined in the Cochrane Handbook for Systematic Reviews of Interventions: 0% to 40% might not be important; 30% to 60% may represent moderate heterogeneity; 50% to 90% may represent substantial heterogeneity; 75% to 100% represents considerable heterogeneity [39]. The planned subgroup analyses were the (1) duration of Tai Chi exercise and (2) styles of Tai Chi exercise. However, we could not carry out all subgroup analyses on the categories outlined in the protocol because there were insufficient data.

A sensitivity analysis was performed to investigate the impact of the risk of bias for sequence generation and allocation concealment in the included trials. We also assessed the quality of evidence for primary and secondary outcomes according to the Grades of Recommendation, Assessment, Development and Evaluation (GRADE) methodology for the risk of bias, inconsistency, indirectness, imprecision, and publication bias [40]. The GRADE approach specifies four levels of quality, the highest of which is for randomized trial evidence. It can be downgraded for moderate, low, or very low-quality evidence.

## 3. Results

### 3.1. Search Results

We retrieved 932 records from the search of the electronic databases and screened a total of 323 records (Figure 1). We reviewed full copies of 15 reports and identified 12 with the potential for meeting the review inclusion criteria. We excluded three records because they included combined mindfulness–body exercise therapies [41,42] or showed no outcome of interest [43].

### 3.2. Characteristics of Included Studies

We identified and included in this review eight trials with 12 records and a total of 822 participants [44,45,46,47,48,49,50,51] (Table 1). In this review, we identified four trials that had published their subsequent follow-up data. One trial reported depressive symptoms among centrally obese adults in Brisbane, Australia [49], with a subsequent follow-up paper in 2019 reporting on QOL [52]. This trial is referred to as Liu et al. (2015). Another trial by Liao et al. (2018) reported on depressive symptoms with follow-up data in the same year and the effects on QOL among older individuals with mild to moderate depressive symptoms in Sichuan Province, China. We also included a pilot study done among the Chinese community in Boston, Massachusetts, published in 2012 [50] with five years of follow-up data that reported on depressive symptoms and QOL [53]. This trial is referenced as Yeung et al. (2012). One trial in Hong Kong reported on the effects of Tai Chi among older Chinese patients with depressive disorder [46], with a follow-up paper in 2008 [54]. This trial is referred to as Chou et al. (2004).

Three trials were conducted in China [47,48,51], another trial was conducted in Australia [49], three trials were conducted in the Unites States [44,45,50], and one trial was conducted in Hong Kong [46]. One trial recruited participants from education centers [51], one trial recruited participants from healthcare settings [46], three trials recruited participants from communities [44,47,48], and three trials recruited participants from healthcare settings and communities [45,49,50].

The durations of the Tai Chi interventions varied at 24 weeks [49], 12 weeks [45,46,47,50], 10 weeks [44], and eight weeks [51]. Participants in one trial were given the antidepressant escitalopram for a certain duration before the intervention [44]. Three trials included participants with non-specific antidepressants [45,49,50]. In four trials, the participants were not given any antidepressant prior to the Tai Chi intervention [46,47,48,51]. For the Tai Chi styles, we identified three trials using Yang’s style [46,47,50], one using KaiMai’s style [49], one using a combination of 24-form and 48-form Tai Chi [48]. and two trials using the Tai Chi Chih (TCC) protocol [44,45]. The TCC is a brief standardized version of Tai Chi adapted from a manual protocol [55]. Although the details of each style are beyond the scope of the present study, 24-form Tai Chi involves 24 forms based on the Yang style, in which 88 forms are condensed to 24, while 48-form Tai Chi was created by combining four major styles: Chen, Yang, Wu, and Sun [56]. One trial did not specify the style of Tai Chi exercise [51].

**Table 1 ijerph-20-02828-t001:** Characteristics of the included trials.

Author (Year)	Publication	Country	Numbers of Participants Included	Age/Mean Age (SD)	Study Duration (Weeks)	Outcome (Measurement Tools)
Liao et al. (2018)	[47]	China	Intervention: 58 Control: 56	Intervention: 71.84 (7.297) Control: 71.75 (8.201)	12	1. Depressive level - GDS
	[57]			Intervention: 71.72 (7.331) Control: 71.87 (8.002)		1. QOL (Mental health) - WHOQOL-BREF 2. QOL (Physical function) - WHOQOL-BREF
Zhang et al. (2018)	[51]	China	Intervention: 32 Control: 32	Intervention: 18.41 (2.01) Control: 18.41 (2.01)	8	1. Depressive level - PHQ-9 2. QOL (Mental health) - MAAS
Yeung et al. (2012)	[50]	United States	Intervention: 26 Control: 13	Intervention: 54 (12) Control: 58 (7)	12	1. Depressive level - HAM-D - CGI-S 2. QOL (Mental health) - Q-LES-Q-SF
	[53]		Intervention: 23 Education: 22 Waitlist: 22	Intervention: 53 (14) Education: 55 (9) Waitlist: 55 (15)		1. Depressive symptoms - BDI - HAM-D - CGI-S 2. QOL (Mental health) - SF-36 - MAAS 3. QOL (Physical function) - SF-36
Lavretsky et al. (2011)	[44]	United States	Intervention: 36 Control: 37	Intervention: 69.1 (7.0) Control: 72.0 (7.4)	10	1. Depressive symptoms - HAM-D 2. QOL (Physical function) - SF-36
Lavretsky et al. (2021)	[45]	United States	Intervention: 89 Control: 89	Intervention: 69.2 (6.9) Control: 69.4 (6.2)	12	1. Depressive level - HAM-D - GDS 2. QOL (Mental health) - CD-RISC - SF-36
Chou et al. (2004)	[46]	Hong Kong	Intervention: 7 Control: 7	Intervention: 72.6 (4.2) Control: 72.6 (4.2)	12	1. Depressive level - CES-D
	[54]					1. Depressive level - CES-D
Liu et al. (2015)	[49]	Australia	Intervention: 106 Control: 107	Intervention: 52 (12) Control: 53 (11)	24	1. Depressive level - CES-D - DASS
	[52]					1. QOL (Mental health) - SF-36 2. QOL (Physical function) - SF-36
Liu et al. (2018)	[48]	China	Intervention: 30 Control: 30	Intervention: 60.9 (4.28) Control: 61.72 (3.54)	24	1. Depressive level - GDS

Note: CES-D: Chinese version of the Center for Epidemiological Studies Depression Scale; BDI: Beck’s Depression Inventory; CGI-S: Clinical Global Impression–Severity; CD-RISC: 25-item Connor–Davidson Resilience Scale; DASS: Depression Anxiety Stress Scale; GDS: Geriatric Depression Scale; HAM-D: Hamilton Rating Scale for Depression; MAAS: Mindful Attention and Awareness Scale; PHQ-9: Nine-Item Patient Health Questionnaire Depression Scale; Q-LES-Q-SF: Quality of Life Enjoyment and Satisfaction Questionnaire; QOL: Quality of life; SF-36: 36-item Short Form Health Survey questionnaire; WHOQOL-BREF: World Health Organization Quality of Life Instrument.

### 3.3. Outcomes

The primary outcome, the depressive level, was reported in eight trials [44,45,46,47,48,49,50,51]. The secondary outcomes measured were mental-health-related QOL [45,47,49,50,51], physical-health-related QOL [44,47,49,50], and anxiety [44,49].

The depressive scales used were the Geriatric Depression Scale (GDS), the Nine-Item Patient Health Questionnaire Depression Scale (PHQ-9), the Hamilton Rating Scale for Depression (HAM-D), the Clinician’s Global Impression of Patient-Severity (CGI-S), the Chinese version of the Centre for Epidemiological Studies Depression Scale (CES-D), the Depression Anxiety Stress Scale (DASS), and Beck’s Depression Inventory (BDI). Three trials reported depressive levels measured using the GDS [45,47,48], a self-rated scale for older individuals. It consists of 30 items with yes or no responses to measure a person’s emotional state over the previous week. The total scores range from 0 to 30, with 0–10 indicating no depression, 11–20 indicating mild depression, 21–25 indicating moderate depression, and 26–30 indicating severe depression. One trial reported using the PHQ-9 [51], which comprises nine items scored on a five-point scale ranging from 0 to 27, with higher scores indicating more severe cases of depression. Three trials used the HAM-D [44,45,50]; for the majority of the questions on that scale, the scores range from 0 to 4, with 4 indicating more acute signs of depression. Several questions contain answer ranges that do not exceed 2 or 3, with higher scores again indicating more severe depression. One trial used the CGI-S [50], which measures the patient’s current condition, as determined by the clinician, on a scale from 1 to 7; the higher the score, the more severe the depression. One trial used the BDI [50]; its 21 items include depressive symptoms and attitudes reflective of its severity. Scores ranging from 0 to 9 indicate no or mild depression, scores from 10 to 18 indicate mild to moderate depression, scores from 19 to 29 indicate moderate to severe depression, and scores from 30 to 63 indicate severe depression. Two trials used the Chinese version of the CES-D, a 20-item measure of how often over the previous week patients have experienced depressive symptoms as assessed by the caregiver [46,49]. The response options range from 0 to 3 for each item (0 = rarely or none of the time; 1 = some or little of the time; 2 = moderately or much of the time; 3 = most or almost all the time). The scores range from 0 to 60, with higher scores reflecting increased severity of depression.

The anxiety scales used were the Hamilton Anxiety Rating Scale (HARS) and the 21-item DASS-21. One trial reported used the HARS [44], which consists of 14 items, including the number of symptoms; each group of symptoms is rated on a scale of 0 to 4, with 4 being the most severe. A higher total score indicates more severe anxiety. One trial used the DASS-21 [49], a well-validated measure of the severity of depression, anxiety, and stress. The total scores range from 0 to 42 for depression, anxiety, and stress, with higher scores indicating more severe depression, anxiety, or stress. For anxiety, scores from 0 to 7 are normal, 8 to 9 indicate mild anxiety, 10 to 14 indicate moderate anxiety, 15 to 20 indicate severe anxiety, and scores above 20 indicate extremely severe anxiety.

Mental-health-related QOL was assessed using several scales, including the 36-Item Short-Form Health Survey questionnaire (SF-36); the Chinese version of the WHO QOL instrument (WHOQOL-BREF); the Mindful Attention Awareness Scale (MAAS); the Quality of Life, Enjoyment, and Satisfaction Questionnaire-Short Form (Q-LES-Q-SF); and the 25-item Connor–Davidson Resilience Scale (CD-RISC). For outcomes of mental-health-related QOL, higher scores indicate a better QOL. Four trials used the SF-36 [44,45,49,50], which uses eight scales, including physical functioning, role limitations for physical reasons, bodily pain, general health, vitality, social functioning, role limitations for emotional reasons, and mental health. Each score ranges between 0 and 100, with higher values representing better QOL. One trial used the Chinese version of WHOQOL-BREF to assess QOL [47]. The WHOQOL-BREF consists of 26 items, including two general items regarding health conditions that would be analyzed independently. The other items evaluate QOL in four main domains: physical health (seven items), psychological (six), social relationships (three), and the environment (eight). Three items are phrased negatively and reverse-scored. The domain scores are computed separately; the raw score of each domain is converted into a transformed score from 0 to 100 [58]. After conversion, the maximum scores for the physical health, psychological, social relationships, and environment domains are 100, with higher scores indicating better QOL. Another trial evaluated quality of life by using the MAAS [51], with higher scores indicating a greater level of mindfulness. One trial used the Q-LES-Q-SF to assess the degrees of enjoyment and satisfaction in various areas of daily functioning [50], with higher scores representing a better QOL. Psychological resilience was assessed in one trial using the 25-item CD-RISC [45]; the total possible scores range from 0 to 100, with higher scores indicating greater resilience.

The physically related QOL was assessed using two scales: the SF-36 and the WHOQOL-BREF. Even though the scales used to assess QOL are general, we retrieved data based solely on the physical domain for this physically related QOL outcome. Higher scores for physically related QOL indicated better QOL. We identified three trials using SF-36 to assess physical functioning [44,49,50], with one using the WHOQOL-BREF physical domain [47].

### 3.4. Risk of Bias in Included Studies

The assessment of the risk of bias is shown in Figure 2 and Figure 3. Figure 2 shows the proportion of studies assessed as low, high, or unclear risk of bias for each indicator. Figure 3 shows the risk of bias indicators for individual studies.

The method of randomization was described for seven trials [1,44,45,47,48,49,50,51]: five used computer-generated randomization [44,45,47,49,50], one used the lottery method [51], and one only noted using a random number generator [48]. The method of randomization was not described in one trial, which was, thus, assessed as having an unclear risk of bias [46].

The method of allocation concealment was described in one trial [45]. The participants were informed that they were in both exercise and wellness education to generate equal expectations, which we considered a low risk of bias. Regarding the fact that the exercise intervention and allocation concealment method were not described in seven trials, we judged these as having a high risk of bias because the enrolled participants were aware of the intervention group and the duration of the trial [44,46,47,48,49,50,51].

Four trials had a low risk for performance bias because the blinding of the participants and personnel was described [44,45,46,51]. The participants were blinded through random assignment into groups [51]; by randomization status [46]; or blinded to the study objectives, randomization status, and outcomes [44,45]. Four trials reported that the participants were not blinded in the trials, although the research personnel were blinded to the randomization status. Because the outcomes were likely to be influenced, we considered these cases to be at high risk for performance bias [47,48,49,50]. The detection bias was judged as a low risk because the blinding of the outcome assessment was described in all eight trials [44,45,46,47,48,49,50,51].

A low risk of bias for incomplete outcome data was assessed for seven trials [44,45,46,47,48,50,51]. Two trials described no missing data [47,48], three trials had dropouts in the intervention group below 20% [44,46,51], two trials carried out an intention-to-treat analysis by analyzing the participants according to the group to which they were initially assigned [49,50]. One trial reported a high risk of bias for incomplete outcome data because there were missing data for 30% of the intervention group [45]. There was no evidence for selective reporting bias because it was clear that all pre-specified outcomes of interest were reported in the studies, including those that involved published protocols.

### 3.5. Effects of Interventions

Comparison 1: Combinations of Tai Chi and antidepressants versus standard antidepressants.

The first comparison, which involved four trials, compared combinations of Tai Chi and antidepressants with standard antidepressants [44,45,49,50]. As illustrated in Figure 4 and Table 2, the Tai Chi intervention reduced the depressive symptom severity compared to the control group (SMD −0.58; 95% CI −1.13 to −0.03; I^2^ = 92%; *p* = 0.04; four trials; 837 participants; low-quality evidence) [44,45,49,50]. Lavretsky et al. (2011) used one scale, the HAM-D. Lavretsky et al. (2021) used two scales, the HAM-D and GDS. Liu et al. (2015) used two scales, the CES-D and DASS. Yeung et al. (2012) used three scales, the BDI, HAM-D, and CGI-S. A subgroup analysis based on duration and styles of Tai Chi style could not be performed in view of the limited number of trials. The sensitivity analysis by Liu et al. (2015) showed substantial changes in the effect size and CI (SMD −1.59; 95% CI −1.59 to 0.20; I^2^ = 93%; *p* = 0.13) that changed to no difference between the Tai Chi intervention group and the control group. There were no substantial changes in effect sizes or CIs for the other trials.

For secondary outcomes, two trials reported anxiety levels [44,49], three trials reported on mental-health-related QOL [45,49,50], and three trials reported on physically related QOL [44,49,50]. For QOL outcomes, the higher the score, the better the QOL.

As shown in Figure 5 and Table 2, for anxiety level outcomes, two trials reported that the Tai Chi intervention reduced anxiety compared to the control group (SMD −0.45; 95% CI −0.76 to −0.15; I^2^ = 17%; *p* = 0.003; two trials; 219 participants; low-quality evidence) [44,49].

As illustrated in Figure 6 and Table 2, three trials reported that standard antidepressants increased the mental-health-related QOL compared to the Tai Chi intervention (SMD 0.28; 95% CI −0.08 to 0.63; I^2^ = 71%; *p* = 0.12; three trials; 519 participants; low-quality evidence) [45,49,50]. Lavretsky et al. (2021) used two scales, the CD-RISC and SF-36. Yeung et al. (2012) used two scales, the SF-36 and *MAAS*. The sensitivity analysis performed by Liu (2015) showed substantial changes in effect sizes and CI values (SMD 0.40; 95% CI 0.08 to 0.72; I^2^ = 44%; *p* = 0.01) that indicated that Tai Chi increased the mental-health-related QOL compared to the control group. There were no substantial changes in effect sizes or CIs for the other trials.

As shown in Figure 7 and Table 2, three trials reported that Tai Chi led to improved physically related QOL when compared to the standard antidepressant group (SMD 0.55; 95% CI 0.33 to 0.78; I^2^ = 0%; *p* < 0.001; three trials; 318 participants; low-quality evidence) [44,49,50]. A sensitivity analysis with all trials showed no substantial changes in effect sizes or CIs.

Comparison 2: Tai Chi versus no intervention.

The second comparison, encompassing four trials, compared Tai Chi with no intervention in terms of the primary outcome and the secondary outcome of mental-health-related QOL [46,47,48,51]. No trials measured anxiety levels or physically related QOL in this comparison.

As illustrated in Figure 8 and Table 3, the review reported that the Tai Chi interventions reduced the depressive symptom severity when compared to the control group (SMD −1.32; 95% CI −2.11 to −0.52; I^2^ = 85%; *p* = 0.001; four trials; 243 participants; very low-quality evidence) [46,47,48,51]. A subgroup analysis based on duration and styles of Tai Chi could not be performed because of the limited number of trials. A sensitivity analysis with all trials showed no changes in effect sizes.

As shown in Figure 9 and Table 3, Tai Chi increased the mental-health-related QOL compared to the control group (SMD 1.01; 95% CI 0.69 to 1.33; I^2^ = 0%; *p* < 0.001; two trials; 263 participants; low-quality evidence) [47,51].

## 4. Discussion

The eight included RCTs were divided into two comparisons addressing several outcomes. Tai Chi exercise reduced depression among patients with depressive symptoms in both comparisons when compared to standard antidepressants and when compared with no exercise. A subgroup analysis based on the duration and style of Tai Chi exercise could not be performed due to the limited number of trials. Hence, this result cannot suggest the styles, intensity, and frequency of Tai Chi exercise suitable for different age group.

Tai Chi exercise as an adjunct to antidepressant therapy reduced the severity of anxiety and improved physically related QOL in patients with depressive symptoms. We were unable to analyze anxiety and physically related QOL in the Tai Chi exercise group due to the limited number of trials. In this review, we also found improvements in mental-health-related QOL outcomes with Tai Chi exercise compared to no exercise; however, the use of standard antidepressants increased the mental-health-related QOL when compared to combined Tai Chi exercise with standard antidepressants. The sensitivity analysis by Liu et al. (2015) on the combination of Tai Chi and antidepressants versus standard antidepressants showed that the significance changed to no difference for the level of depressive symptoms, and a substantial change in the effect estimate was reported for the mental-health-related QOL. These changes affecting these two outcome were due to the large amounts of missing data in our first comparison [49]. The results from this trial could, thus, be biased and should be interpreted with caution.

We performed a comprehensive and extensive literature review to assess the effectiveness of Tai Chi exercise on the mental and physical well-being of patients with depressive symptoms. We included eight randomized trials with 822 participants that included adolescent, adult, and geriatric populations. The control groups varied, with standard antidepressant treatments, health education courses, any form of exercise other than mindfulness-body exercise, and no exercise. The durations of Tai Chi varied between eight and 24 weeks. In the majority of trials, Tai Chi was performed two to three times per week for 45–90 min per session. This evaluation did not, however, assess the intensity of Tai Chi activity in the trials. The results from the trials showed the clinical efficacy of the Tai Chi interventions; however, their small sample sizes may have led to instability in the outcomes. The exclusion of trials published in languages other than English could also limit the relevance of the findings regarding the health benefits.

The overall quality of the evidence for trials comparing combinations of Tai Chi and antidepressants with antidepressants alone was low, whereas the quality levels of the evidence for trials comparing Tai Chi to no exercise ranged from very low to low quality. Due to the absence of concealment and blinding of the individuals, the high variability of the outcomes, and the small sample size, the quality of the evidence was, on the whole, notably suboptimal. Most trials were deemed to have a minimal risk of bias, with the exception of two domains. Seven trials had a high risk for selection bias because allocation concealment could not be performed as Tai Chi exercise is a form of physical exercise [44,46,47,48,49,50,51]. The risk of performance bias was high in four trials because the participants were not blinded in the active intervention, which was likely to have influenced the outcome [47,48,49,50]. We encountered serious inconsistencies in depressive levels in both comparisons and mental-health-related QOL in the first comparison. However, we were unable to carry out a subgroup analysis due to the limited number of similar trials for comparison. Serious imprecision was found in most of our outcomes due to the small sample sizes of the included trials.

We identified some limitations in this review. A subgroup analysis could not be performed due to inadequate number of trials; hence, we unable to explain the heterogeneity between trials. Secondly, the Review Manager 5.4 software (Revman 2020) used in this review did not enable the calculation of effect sizes that could indicate the significance of the research. Some trials examined in this systemic review did not provide details of the Tai Chi practice, such as the styles and intensity involved, which may have influenced the outcome of the trials [44,45,48,51]. The generally low quality of evidence in this review was affected by high risk of bias for allocation concealment and blinding.

We attempted to reduce the publication bias by checking the reference lists of all related studies for further references and by searching multiple databases. A few trials met our preliminary inclusion criteria; however, they were excluded from the meta-analysis because no depressive symptom data were found or they did not assess the use of Tai Chi as a co-intervention with other mindfulness–body exercises. We restricted our review to English published articles only. Thus, we cannot be certain that we were able to locate all trials in this area. Since positive trials are more likely to be published than negative trials, publication bias might have existed in the included studies, and the effect sizes of Tai Chi might have been overestimated.

Tai Chi exercise reduced depression among patients with depressive symptoms receiving antidepressant therapy and in milder forms of depression with no active intervention, such as with antidepressant therapy or other mindfulness–body exercises. The results of this review are also consistent with the previously published reviews on the effects of Tai Chi exercise on improvements in depressive symptoms [16,24,25,26,27,28,29,30,31,35,36,37,38]. Two systematic reviews reported that non-pharmacological therapies could reduce depressive symptoms and should be considered, along with antidepressant therapy, for the treatment of mild to severe depression [16,59]. The evidence of improvements in mental and physical well-being was supported by reductions in depressive symptoms, as well as improvements in both mental- and physical-related QOL indicators. Another review reported improvements in mental and physical health; however, it was conducted among an elderly population [31]. Tai Chi practice also can enhance immunity; hence, it could lead to improved respiratory conditions post COVID-19 infection, which will be beneficial in this COVID-19 era [22,31].

Tai Chi as a form of physical exercise is beneficial for the treatment of depression, as supported by evidence-based articles on non-pharmacological treatment guidelines for depression in Korea. Physical exercise therapy is more efficacious than no treatment for adult patients with mild-to-moderate depression, and its efficacy is similar to that of antidepressant treatment [60]. The evidence of the clinical effect of Tai Chi exercise on depressive symptoms signifies the benefit of its use as a complementary approach to the treatment of depression. However, it is not possible to suggest Tai Chi alone as an intervention for the treatment of depression, unless for milder forms of depression.

## 5. Conclusions

In conclusion, this present systemic review and meta-analysis suggests that Tai Chi interventions have shown improvements in mental and physical well-being as evidenced by reductions in depression and anxiety and improved QOL in patients with depressive symptoms. Tai Chi exercise as a part of a non-pharmacological approach used as an adjunct to antidepressant therapy can reduce depression, along with health education and cognitive behavioral therapy. The identification of definitive outcomes was limited by the inadequate blinding, heterogeneity of the outcomes, and small sample size. Further well-controlled RCTs are recommended with a precision trial design and larger sample sizes. Tai Chi exercise could potentially be used as a new complementary medical approach in the treatment of depression in healthcare settings.

## Figures and Tables

**Figure 1 ijerph-20-02828-f001:**
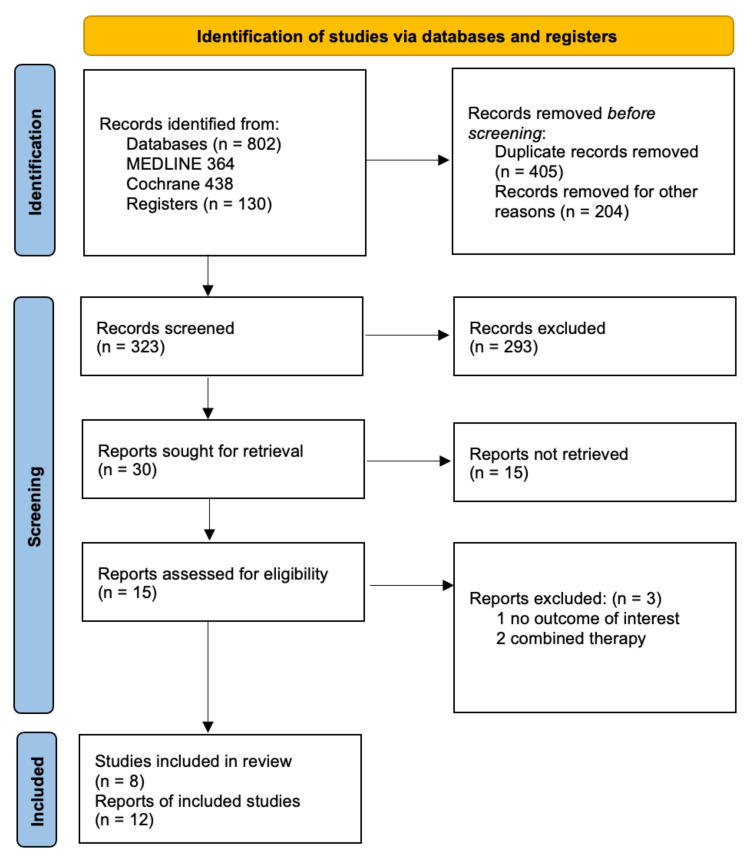
PRISMA flow chart.

**Figure 2 ijerph-20-02828-f002:**
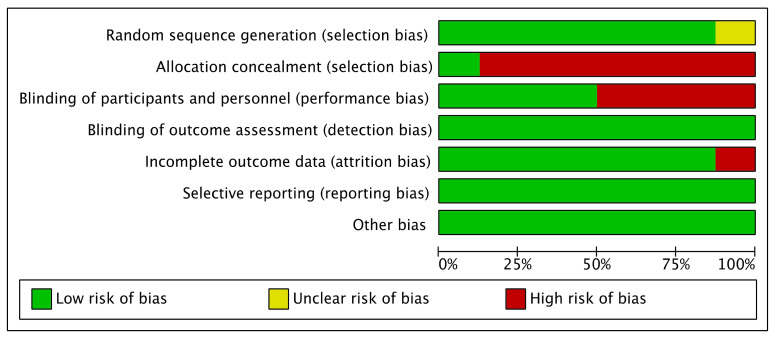
Risk of bias graph, showing the review authors’ judgements of each risk of bias item presented as percentages across all included studies.

**Figure 3 ijerph-20-02828-f003:**
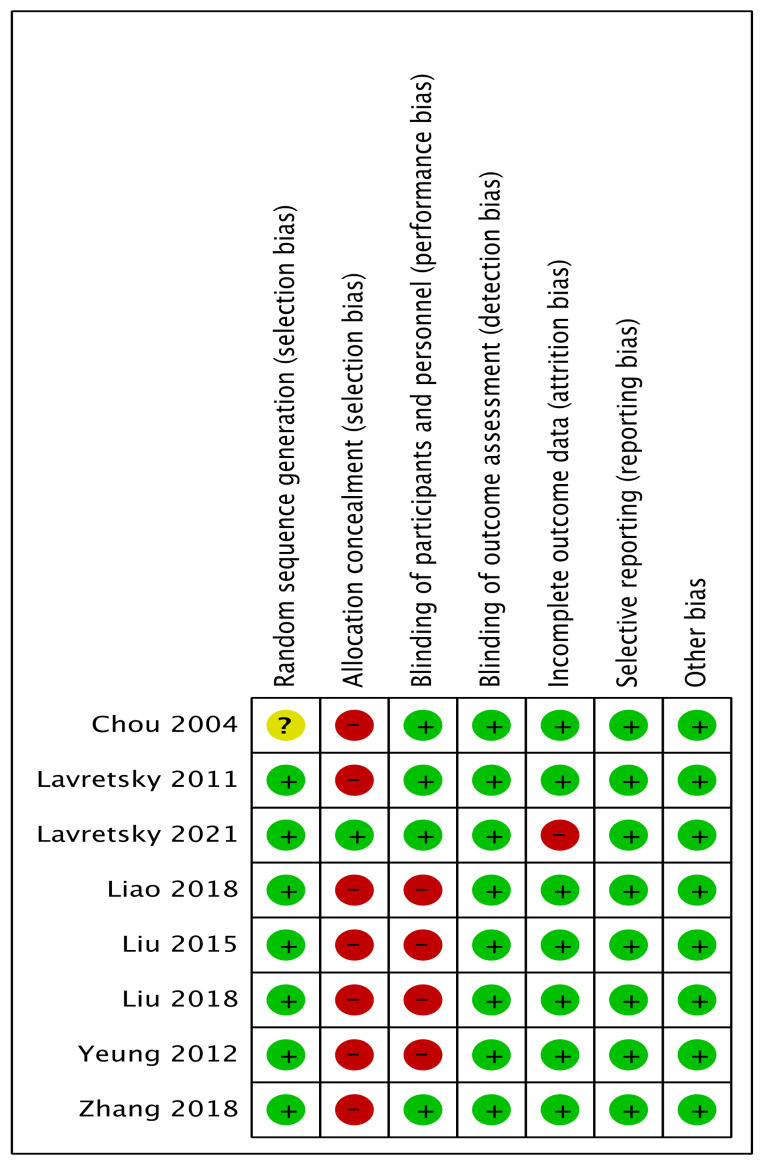
Risk of bias summary, showing the review authors’ judgements of each risk of bias item for each included trial [44,45,46,47,48,49,50,51]. Note: 
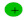
 indicates low risk of bias, 
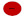
 indicates high risk of bias and 
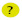
 indicates unclear risk of bias.

**Figure 4 ijerph-20-02828-f004:**
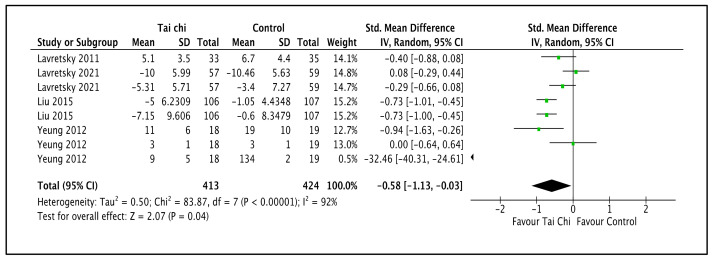
Forest plot provided for comparison of the combinations of Tai Chi and antidepressants versus standard antidepressants for the depressive level outcome [44,45,49,50]. Note: Line indicates confidence interval and the green square indicates effect estimate. Black diamond indicates cumulative effect and its confidence interval.

**Figure 5 ijerph-20-02828-f005:**
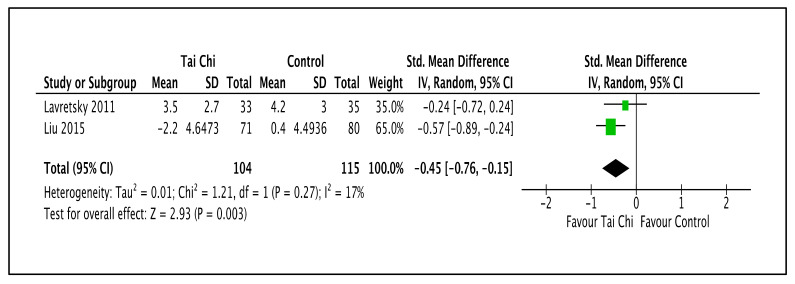
Forest plot provided for comparison of the combination of Tai Chi and antidepressants versus standard antidepressants for anxiety level outcome [44,49]. Note: Line indicates confidence interval and the green square indicates effect estimate. Black diamond indicates cumulative effect and its confidence interval.

**Figure 6 ijerph-20-02828-f006:**
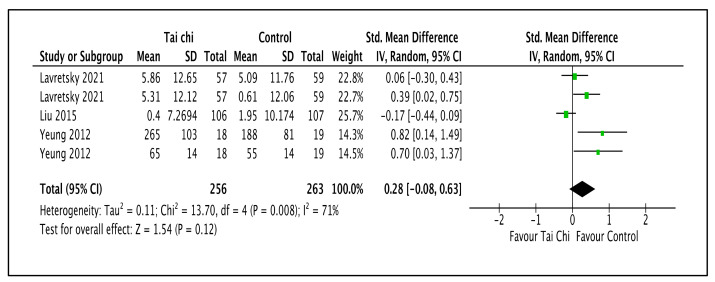
Forest plot shown for comparison of the combinations of Tai Chi and antidepressants versus standard antidepressants for mental-health-related QOL outcome [45,49,50]. Note: Line indicates confidence interval and the green square indicates effect estimate. Black diamond indicates cumulative effect and its confidence interval.

**Figure 7 ijerph-20-02828-f007:**
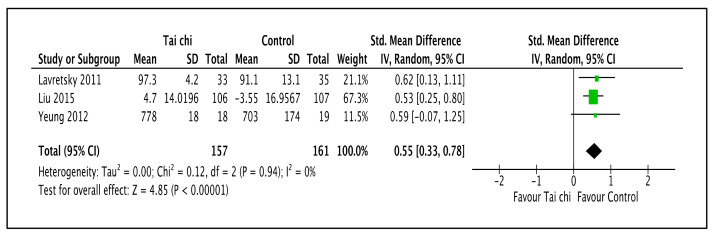
Forest plot shown for comparison of the combinations of Tai Chi and antidepressants versus standard antidepressants for physically related QOL outcome [44,49,50]. Note: Line indicates confidence interval and the green square indicates effect estimate. Black diamond indicates cumulative effect and its confidence interval.

**Figure 8 ijerph-20-02828-f008:**
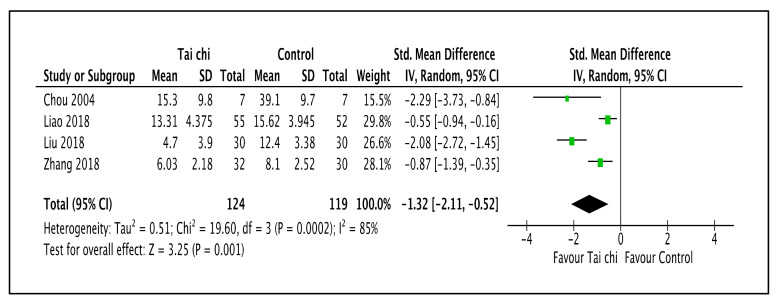
Forest plot shown for comparison of Tai Chi versus no intervention for depressive level outcome [46,47,48,51]. Note: Line indicates confidence interval and the green square indicates effect estimate. Black diamond indicates cumulative effect and its confidence interval.

**Figure 9 ijerph-20-02828-f009:**
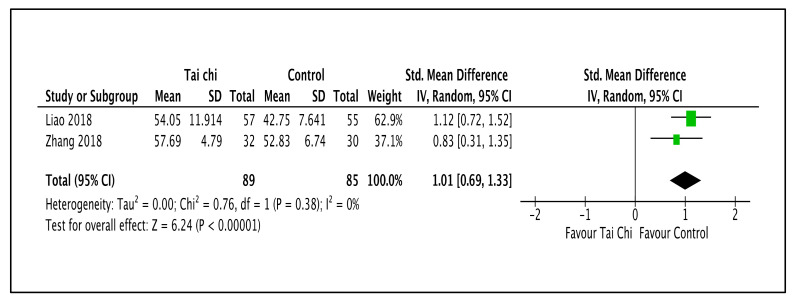
Forest plot shown for comparison of Tai Chi versus no intervention for mental-health-related QOL outcomes [47,51]. Note: Line indicates confidence interval and the green square indicates effect estimate. Black diamond indicates cumulative effect and its confidence interval.

**Table 2 ijerph-20-02828-t002:** Summary of the findings of the GRADE quality assessment for a comparison of combinations Tai Chi and antidepressants versus standard antidepressants.

Certainty Assessment							No of Patients		Effect		Certainty
No of Studies	Study Design	Risk of Bias	Inconsistency	Indirectness	Imprecision	Other Considerations	Tai Chi	Control	Relative (95% CI)	Absolute (95% CI)	
**8**	randomized trials	serious ^a^	serious ^b^	not serious	not serious	none	413	424	-	SMD 0.58 lower (1.13 lower to 0.03 lower)	⨁⨁◯◯ Low
**2**	randomized trials	serious ^a^	not serious	not serious	serious ^c^	none	104	115	-	SMD 0.45 lower (0.76 lower to 0.15 lower)	⨁⨁◯◯ Low
**3**	randomized trials	serious ^a^	serious ^b^	not serious	not serious	none	256	263	-	SMD 0.28 higher (0.08 lower to 0.63 higher)	⨁⨁◯◯ Low
**3**	randomized trials	serious ^a^	not serious	not serious	serious ^c^	none	157	161	-	SMD 0.55 higher (0.33 higher to 0.78 higher)	⨁⨁◯◯ Low

Note: CI: confidence interval; serious^a^: most trials had a lack of allocation concealment and blinding of participants, which might have affected the trial outcomes; serious^b^: substantial heterogeneity of results; serious^c^: small sample size; SMD: standardized mean difference. ⨁/◯ indicates the certainty level.

**Table 3 ijerph-20-02828-t003:** Summary of the findings of the GRADE quality assessment for a comparison of Tai Chi versus no intervention.

Certainty Assessment							No of Patients		Effect		Certainty
No of Studies	Study Design	Risk of Bias	Inconsistency	Indirectness	Imprecision	Other Considerations	Tai Chi	No Intervention	Relative (95% CI)	Absolute (95% CI)	
4	randomized trials	serious ^a^	serious ^b^	not serious	serious ^c^	none	124	119	-	SMD 1.32 lower (2.11 lower to 0.52 lower)	⨁◯◯◯ Very low
2	randomized trials	serious ^a^	not serious	not serious	serious ^c^	none	89	85	-	SMD 1.01 higher (0.69 higher to 1.33 higher)	⨁⨁◯◯ Low

Note: CI: confidence interval; serious ^a^: most trials had a lack of allocation concealment and blinding of participants, which might have affected the trial outcomes; serious^b^: substantial heterogeneity of the results; serious ^c^: small sample size; SMD: standardized mean difference. ⨁/◯ indicates the certainty level.

## Data Availability

The original contributions presented in the study are included in the article materials. Further inquiries can be directed to the corresponding author.

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
