# Peer review of "Tai Chi Exercise for Mental and Physical Well-Being in Patients with Depressive Symptoms: A Systematic Review and Meta-Analysis"

_ijerph, 2023, doi:10.3390/ijerph20042828_

Round 1

Reviewer 1 Report (Previous Reviewer 2)

The authors have responded to the review satisfactorily, although some issues with written expression compromise the clarity of the information provided. 

Author Response

Manuscript Revised Review
IJERPH 2169788_R (prev 2002674)
Title Tai chi exercise for mental and physical well-being in patients with depressive symptoms: a systematic review and meta-analysis 

Thank you for giving us a chance to have another revision and submit our manuscript in correction; we really appreciated the reviewer's comments. Below is our response to the reviewers` comments.

General

The authors respond that they use "physical well-being" to refer to the physical aspect of quality of life. They could therefore use the term quality of life in the title and manuscript text to more accurately reflect study outcomes eg, depression, anxiety, and quality of life (v.s mental and physical well-being). Physical well-being a is very generic thatwhich could be understood to mean physical indicators eg functioning etc.

Respond:

We write as 'mental and physical well-being' to give a reader a clearer understanding of this research topic. We will describe QOL indicator as part of mental and physical well-being in our discussion part.  

Abstract

Some wording has been revised as requested, and the period for included studies and method used to assess study quality has been added.

1) The two comparisons (lines 21-23) are still unclear to me (though I understand them from the results text), and I think this is an issue of written expression. The first comparison describe tai chi as an intervention, but both comparisons do that. The results indicate that the included studies compared (1) tai chi vs no intervention and (2) tai chi + antidepressants vs antidepressants.

The eight trials were divided into two comparisons: assessing Tai Chi as intervention and as an adjunct to the standard antidepressant. (Lines 21-23)

Respond: we made correction as stated below

Corrections done:

The eight trials were divided into two comparisons: (1) combination Tai Chi and antidepressants versus standard antidepressants and (2)  Tai Chi versus no intervention.

Lines: 21-23

2) There are some issues with written expression eg (line 12) have should be has, practise (a verb) should be practice (the noun), "the" is missing before treatment.

Respond: we made correction as stated below

Tai Chi is a mindfulness-body practise that have physiological and psychosocial benefits to be integrated into the prevention and rehabilitation of various medical conditions, however the effectiveness of Tai Chi in treatment of depression remain unclear. (Line 12)

Correction done:

Tai Chi is a mindfulness-body practice that has physiological and psychosocial benefits to be integrated into the prevention and rehabilitation of various medical conditions, however the effectiveness of Tai Chi in the treatment of depression remain unclear.

Lines: 12

3) Lines 17-18 could be revised eg studies were required to be a RCT, focus on people with depression and no other medical conditions, and included both adolescent and adult samples. The information on line 18 is not required.

Respond: we made correction as stated below

Randomized controlled trials that involved participants with depressive symptoms regardless of the scales used either, subjective or objective assessment, age, gender and ethnicity were included. (Lines 17-18)

Correction done: Trials included to be a RCT, that involved people with depression with no other medical conditions, and included both adolescents and adult samples.

Lines: 17-18

Introduction

The authors have extended the rationale for the current study and described other reviews of Tai chi. The discussion of previous research studies limited to older adults, and studies with people who have both medical conditions and depression is relevant as it highlights gaps to be addressed in the current research i.e. across the lifespan and among people with depression but no medical condition.

1) There may be a typographical error in lines 85 to 88 where the authors describe Tai chi having a significant impact upon psychological responses: should this refer to people across a variety of chronic conditions (vs as well as?)

Respond: we made correction as stated below

Lines 85-88: Tai Chi also has been shown to have a significant impact on psychological responses as well as a variety of chronic conditions such as cardiovascular and respiratory function, musculoskeletal, hypertension, endochrine and immune systems [27, 28].

Correction done:

Tai Chi also has been shown to have a significant impact on  depression, anxiety and physical well-being among people with variety of chronic conditions [27, 28].

Lines: 91-93

2) Information about Tai chi improving physical functional status, insomnia and chronic pain (lies 92-94) could indicate that these conditions are common among people with depression, and/or components of quality of life, to provide some relevance to the current study.

Lines 92-93: Besides improving the depression, Tai Chi also help to improve physical functional status [36], insomnia [37] and chronic pain [38].

Respond: we made correction as stated below

Tai Chi exercise also had shown better QOL by improving physical functional status [36], insomnia [37] and chronic pain [38], in which these conditions common among people with depression.

Lines: 97-99

3) Some of the cited research would benefit from some more specificity to describe the outcome variables and participant samples in order to indicate the innovation/value of the current study e.g., "assessed the effect of tai chi on medical conditions" (line 97) is unclear: were studies aiming to reduce symptoms of medical conditions or depressive symptoms among people with medical conditions (or both)?

Respond: we made correction as stated below

Few systematic reviews reported reduction of medical-related symptoms in people with cardiovascular-related illness [31, 35, 36], fibromyalgia [37], and multiple sclerosis [38]. Those reviews assessed the effect of Tai Chi aiming to reduce symptoms of medical conditions and depressive symptoms, hence their results were more complex and unclear. Several studies assessed depressive symptoms as secondary outcome, hence the participants included can be participants without baseline depression. Most of trials examined depressive symptoms as secondary outcome of study and the participants involved has underlying medical conditions that can be debilitating physically and leads to chronic pain which can influence severity of the depressive outcome.  The physiological and psychosocial effect of Tai Chi is beneficial for the prevention and rehabilitation of various medical conditions, however the effectiveness of Tai Chi in treatment of depression among patient with depressive symptoms with no other medical conditions remain unclear.

Lines: 99-188

4) How does this study differ from refs 24-26 (line 85) which seem to have the same broad description as the current study?

Lines 83-85: Several studies have reported the effectiveness of Tai Chi exercise as part of the non-pharmacological approach to treat patients with depression, and it has been associated with improvement in mental health and QOL [24-26].

Respond: we made correction as stated below

In these reviews (refs 24-26), they studied the depression in general healthy population, whereas in our review, the studies included if the participants in the studies reported depressive symptoms regardless of the scales as mentioned in our study eligibility.

Ref 24: this review assessing the depression among healthy individuals

Ref 25: this review assessing the depression among healthy older adults

Ref 26: this review assessing the depression among healthy individual

Corrections made: We edited our sentences to give better understanding

Several trials have reported the effectiveness of Tai Chi exercise as part of the non-pharmacological approach to treat patients with depression, and it has been associated with improvement in mental health and QOL among healthy general population [24-26].

Lines: 89-91

5) The information on line 104 can be extended slightly to indicate this study focused on people with depressive symptoms and no other specific medical conditions.

Lines 104-106: This present review aimed to overcome some limitations of existing reviews by analysing the effect of Tai Chi on mental and physical well-being in patient with depressive symptoms.

Respond: we made correction as stated below

This present review aimed to overcome some limitations of existing reviews by analysing the effect of Tai Chi exercise on mental and physical well-being in patient with depressive symptoms and no other specific medical conditions.

Lines: 163-165

Materials and Methods

An overview of study search terms and inclusion criteria is now provided. It is now clarified that studies were excluded if samples included people with chronic physical illness and if interventions other than tai chi (apart from antidepressants) were used.

1) The authors have clarified that participant depression could be established by a variety of means. There are some issues with written expression (eg lines 125-127) which compromises the clarity of this information.

Lines 125-127: The participants that reported depressive symptoms regardless of the scales used either, subjective or objective assessment, age, gender and ethnicity were included. Mind-body exercise other than Tai Chi were excluded.

Respond: we made correction as stated below

The participants that reported depressive symptoms based on subjective or objective assessment, with regardless of age, gender and ethnicity were included. Participants with medical illness were excluded in this review. Trials that included Tai Chi as intervention or co-intervention to other mindfulness-body exercise were excluded.

Lines: 185-188

2) The authors' response on how follow up studies or multiple publications from the same study were managed does not address my question. In these cases, which results were used in the analyses?

Lines 195-205: In this review, we identified four trials that had published their subsequent follow-up data. One trial reported depressive symptoms among centrally obese adults in Brisbane, Australia [49], with a subsequent follow-up paper in 2019 reporting on QOL [52]. This trial is referred to as Liu et al. (2015). A trial by Liao et al. (2018) reported on depressive symptoms with follow-up data in the same year on QOL among older persons with mild to moderate depressive symptoms in Sichuan Province, China. We also included a pilot study done among the Chinese community in Boston, Massachusetts, published in 2012 [50] with five years of follow-up data that reported on depressive symptoms and QOL [53]. This trial is referenced as Yeung et al. (2012). One trial in Hong Kong reported on the effects of tai chi among older Chinese patients with depressive disorder [46], with a follow-up paper in 2008 [54]. This trial is referred to as Chou et al. (2004).

Respond:

1) Liu et al (2015): first report on depression outcome with subsequent report on QOL

Depression outcome: We using the data in Liu et al (2015)

QOL: We using the data in Liu et al (2019) because the follow up data analysed on QOL only

2) Liao et al (2018) : first report on depression outcome with subsequent report on QOL within same year

3) Yeung et al (2012)

For depression and QOL outcome: We using the data in Yeung et al 2017 because the latest data assessing both outcome

4) Chou et al (2004)

Depression outcome: We using the data in in Chou et al (2004) because the report in 2008 is an analysis using multiple regression analysis based on data extracted in 2004

Results

1) Please check lines 186, 195, 212 for use of characters instead of words.

Line 186:  As illustrated in Error! Reference source not found., we retrieved 932 records from the search of the electronic databases and screened a total of 323 records.

Line 195: We identified and have included in this review eight trials with 12 records and a total of 822 participants [44-51] as presented in Error! Reference source not found..

Line 212: ...... settings and communities [45, 49, 50] presented in Error! Reference source not found. for full details.

Respond: we made correction as stated below

Line 186:  We retrieved 932 records from the search of the electronic databases and screened a total of 323 records (Figure 1)

Line 195: We identified and have included in this review eight trials with 12 records and a total of 822 participants [44-51] ( Error! Reference source not found.).

Line 212: ...... settings and communities [45, 49, 50] ( Error! Reference source not found.).

Discussion and Conclusions

1) The authors' response that the statistical software used did not enable calculation of effect sizes could be acknowledged as a study limitation

Respond: We had added as part of limitation in this review

We identified some limitation in this review. The subgroup analysis cannot be performed due to inadequate number of trials, hence we unable to explain the heterogeneity between trials. Secondly, the Review Manager 5.4 software (Revman 2020) used in this review did not enable calculation of effect sizes that can indicate significant of research outcome.

Lines: 535-536

2) My initial review indicated that the authors could discuss further the different results and underlying mechanisms. The authors differentiated between studies which used tai chi alone and those which used it with antidepressants, however this is not considered in the discussion

Respond: some addition was added following your suggestions

The eight included RCTs were divided into two comparisons addressing several outcomes. Tai Chi exercise reduced depression among patients with depressive symptoms in both comparisons; when compared to standard antidepressants and when compared with no exercise. Subgroup analysis based on duration and style of Tai Chi exercise could not be performed due to the limited number of trials. Hence, this result cannot suggest the styles, intensity and frequency of Tai Chi exercise suitable for different age group.

Tai Chi exercise as an adjunct to antidepressants therapy reduced the severity of anxiety and improved physically- related QOL in patients with depressive symptoms. We unable to analyse anxiety and physically-related QOL in Tai Chi exercise group due to limited trials. This review also reported improvement in mental-related QOL outcome in Tai Chi exercise compared to no exercise, however standard antidepressants increased the mental health-related QOL when compared to combined Tai Chi exercise with standard antidepressants. The sensitivity analysis in Liu et al. (2015) for combination of Tai Chi and antidepressants versus standard antidepressants had shown the significance changed to no difference for level of depressive symptoms and a substantial change in the effect estimate reported for mental health-related QOL. This changes that affecting these two outcome due to high amount of missing data in our first comparison [49]. The results from this trial could thus be biased and should be interpreted with caution.

Lines: 489 - 506

3) Why would tai chi + antidepressant but not tai chi alone improve physical QoL? Why would tai chi alone but not tai chi + antidepressants improve mental QoL? I do not agree with the response (not included in the manuscript) that QoL is not a clinical indicator. As indicated by the authors, quality of life includes physical and psychological health which would be expected to be responsive to Tai chi. I acknowledge the authors' response that the social and environmental aspects of QoL may be less amenable to change from Tai chi. Some explanation of the difference in the results of the two comparison would strengthen the discussion.

Respond:  

We agreed from previous reviews on QOL outcomes had shown that improvement in QOL, however our review had some limitation due to limited number of trials in between the comparisons groups

For mental related QOL in first comparison: Standard antidepressants increased the mental health-related QOL compared to Tai Chi intervention. In second comparison: Tai Chi increased the mental health-related QOL compared to the control group.

We corrected in our discussion that one trial had affected the mental-related QOL outcome due to high amount of missing data in our first comparison.

We cannot assess physical related QOL in Tai Chi vs No intervention because no trial assessing physical related QOL in 2nd comparison. Otherwise, Tai Chi improved physically-related QOL in those patient on antidepressants.

4) There is no mention of tai chi in the context of antidepressant use in the conclusions.

Lines 482- 489: In conclusion, this present systemic review and meta-analysis suggest that Tai Chi intervention had shown improvement in mental and physical well-being as evidenced by reduction of depression and anxiety, and improved QOL of patients with depressive symptoms. Definitive outcomes were limited by inadequate blinding, heterogeneity outcome, and small sample size. Further well-controlled RCTs recommended with precision on trial design and larger sample sizes. Tai Chi exercise potentially to be used as new complementary medical approach in the treatment of depression to be used in healthcare settings.

Respond: we made correction as suggested

In conclusion, this present systemic review and meta-analysis suggest that Tai Chi intervention had shown improvement in mental and physical well-being as evidenced by reduction of depression and anxiety, and improved QOL of patients with depressive symptoms. Tai Chi exercise as a part of non-pharmacological approach used as adjunct to antidepressant therapy can reduce depression, other than health education and cognitive behavioural therapy. Definitive outcomes were limited by inadequate blinding, heterogeneity outcome, and small sample size. Further well-controlled RCTs recommended with precision on trial design and larger sample sizes. Tai Chi exercise potentially to be used as new complementary medical approach in the treatment of depression to be used in healthcare settings.

** There are some issues with written expression in the discussion and conclusion.

Reviewer 2 Report (Previous Reviewer 1)

I have carefully reviewed the revised manuscript. I think the manuscript is acceptable.

Author Response

REVISION AND RESUBMIT

Previous Manuscript ID: IJERPH 2169788_R (prev 2002674)

Title: Tai Chi Exercise for Mental and Physical Well-Being in Patients with Depressive Symptoms: A Systematic Review and Meta-Analysis

Thank you again for giving us the chance to have a second revision and resubmit our manuscript. We appreciated the comments from both reviewers. The manuscript attached was in the track changes version to submit to reviewer 1. The corrections and add-on texts were highlighted in yellow in the manuscript based on reviewer 1's comments. We have already sent our manuscript to Scrivendi for grammar review.

Yours

Dr Siti Suhaila Mohd Yusoff

This manuscript is a resubmission of an earlier submission. The following is a list of the peer review reports and author responses from that submission.

Round 1

Reviewer 1 Report

1. Meta-analysis is used as a scientific method to collect and integrate evidence from all relevant studies on a research question to address controversial or inconclusive parties. However, I did not see this section in your introduction and the issue was not raised for good reasons.

2. Please add to the introduction the current research progress of meta-analysis in this theme.

3. Please describe in detail the inclusion and exclusion criteria of the literature.

4. Please provide a detailed search strategy so that the results are reproducible. The search strategy report specification should include: sampling strategy, study type, access, inclusion year (start date), restrictions, inclusion and exclusion criteria, search form used, and electronic resources.

5.Wrong reference citation source and wrong layout do not seem to be a friendly problem, line 173-174 and line 182-183.

6. Please unify the expression of Tai Chi.

Author Response

Thank you for giving us chances to revise and resubmit our manuscript, we really appreciated the comments from reviewer. Below is our response to the reviewers` comments.

Author Response

please see in the attachment

Reviewer 3 Report

This manuscript entitled “Tai Chi Exercise for Mental and Physical Well-Being in Patients with Depressive Symptoms: A Systematic Review and Meta-Analysis” primarily aimed to explore the role of Tai Chi exercise in the improvement of psychological well-being and quality of life in depressive patients. The authors bring an interesting study, but there are still some problems that cannot up this review to a publishing level. Suggestions are listed in the specific comments below.

Specific comments:

1.     In the abstract part, line 11-12, “Tai chi exercise was associated with improvement in psychological well-being and quality of life in the general population.” What’s the link between general population and patients with depressive symptoms? Why you mentioned the general population, but systematically review the patients with depressive symptoms? Please improve the logic.

2.     In the abstract part, please add more specific inclusion criteria of the publications.

3.     In the introduction part, line 59-61, “Several studies have reported the effectiveness of tai chi exercise as part of the non-pharmacological approach to treating patients with depression, and it has been associated with improvement in psychological well-being and QOL among the general population.” Please cite more references here.

4.     In the introduction part, last paragraph, the necessity of the manuscript is not well presented, please revise.

5.     In the results part, line 173-174, “see。 for full details.” Please revise this mistake.

6.     Please add some recently studies in the discussion, such as:

Can Yoga, Qigong, and Tai Chi Breathing Work Support the Psycho-Immune Homeostasis during and after the COVID-19 Pandemic? A Narrative Review. Healthcare 2022, 10, 1934. https://doi.org/10.3390/healthcare10101934

Effects of Tai Chi Chuan on the Physical and Mental Health of the Elderly: A Systematic Review. Physical Activity and Health, 5(1), 21–27. DOI: http://doi.org/10.5334/paah.70

7.     What are the limitations of this study? Please provide relevant description in the discussion part.

8.     In the conclusion part, please just show the main findings in the conclusion section. (being more "conclusively").

Author Response

please see in the attachment
